# The Effect of the Saudi *Haloxylon ammodendron* Shrub on Silver Nanoparticles: Optimal Biosynthesis, Characterization, Removability of Mercury Ions, Antimicrobial and Anticancer Activities

**Ahmed N. Al-Hakimi** [1,2,*] , **Tahani M. Alresheedi** [1] **and Reema A. Albarrak** [1]

1 Department of Chemistry, College of Science, Qassim University, Buraidah 51452, Saudi Arabia; tm.alrashidi@qu.edu.sa (T.M.A.); 421200486@qu.edu.sa (R.A.A.)
2 Department of Chemistry, Faculty of Science, Ibb University, Ibb 70270, Yemen
* Correspondence: a.alhakimi@qu.edu.sa or alhakemi10@yahoo.com

**Abstract:** This research provides a sustainable way to treat water by removing heavy metal hazards (mercury ion) and biological pollutants (several strains of bacteria and fungi) through the eco-friendly synthesis of silver nanoparticles using the ethanol extract of the Saudi *Haloxylon ammodendron* shrub, which is planted in the Qassim desert. Further, this work confirms that these nanoparticles could be used as anticancer materials. The optimization factors of the biosynthesis of silver nanoparticles were studied and obtained (volume ratio = 1:2, pH = 7.5, and temperature = 60 °C). The scanning electron microscope micrographs showed the spherical shape and the huge numbers of silver nanoparticles accumulated, while X-ray diffraction measurements gave the crystal size of these nanoparticles in the range of 10.64 nm. The application findings of these biofabricated silver nanoparticles demonstrated effective detection and removal of different concentrations of mercury ions (0–2500 ppm) from the polluted aqueous solutions. The work revealed that *Haloxylon ammodendron* extract enhanced the antibacterial and antifungal activities of silver nanoparticles against different strains of bacteria and fungi. As well, the anticancer activity examinations of these nanoparticles and the extract showed good and reasonable results.

**Keywords:** *Haloxylon ammodendron*; Ag nanoparticles; $Hg^{2+}$ removal; anticancer; antibacterial; antifungal

## 1. Introduction

With the industrial and economic development in the Kingdom of Saudi Arabia and climatic changes, the percentage of environmental pollution in the country is increasing [1–3]. By the way, this is in line with the 2030 vision of the Kingdom of Saudi Arabia towards a clean environment free of pollutants [4]. Universities devote all of their scientific and research resources to achieving this vision by utilizing cutting-edge, inexpensive scientific methods [5,6]. We will talk here briefly about the most important pollutants and then the techniques used to remove them.

The detrimental effects of heavy metals and their numerous harms, such as cancer, are recognized as one of the most crucial topics that must be addressed when discussing environmental contaminants linked to industrial growth [7–9].

Mercury is among the heavy chemical elements and may be found in nature in a variety of forms and is employed in a variety of industries. Despite this, the need to reduce its uses and innovate methods to detect it, as well as ways to get rid of it, has become more of an interest to researchers in the field of the environment [10–12].

Research has shown that many traditional techniques can be used in the treatment and disposal of mercury, for example, membrane filtration, soil utilization, sedimentation, adsorption, the electrocoagulation process, etc., but most of these techniques are time-consuming and financially costly and therefore useless [13–15].

On the other hand, climate change and the growth of epidemics and diseases are also a result of biological pollution, which is another negative side effect of industrial development. This particularly occurs when drinking water is contaminated with different microorganisms [16–18].

Recently, nanotechnology has become one of the most common and important modern methods for dealing with different kinds of pollution, such as heavy metals or microbial contamination [19,20]. Researchers have become interested in this technology because of the special and superior characteristics of nanomaterials in the field of pollution treatment in terms of cost, simplicity of preparation, controllability, and other relevant aspects [21–24].

Recently, research articles have been concerned with one of the most interesting global issues: wastewater treatment using different techniques [25,26]. In this regard, many modern scientific studies have shown the high efficiency of silver nanoparticles in the simple colorimetric detection of mercury ions and explained the mechanism of removing this ion from drinking water in an easy way [27,28].

By the way, there are also many scientific articles that have proven the ability of silver nanoparticles to get rid of microbes that cause diseases and pollution, depending on the characteristics of these nanoparticles, such as the size and quality of the catalysts used in the preparation, the pH of the reaction medium, etc. [29,30].

The effectiveness of nanoparticles in treating microbial infections has been demonstrated by the large number of recent scientific studies that have concentrated on employing nanoparticles to target microbes instead of antibiotics. This prompted scientists to develop several preparation methods to improve the biological activity of these nanoparticles as anticancer materials. The extraordinary properties of silver nanoparticles in various applications have piqued researchers' interest in these sorts of nanoparticles for their potential antimicrobial and anticancer properties [30–33].

This research work presented the bio-manufacturing of silver nanoparticles using an ethanolic extract of *Haloxylon ammodendron* shrub under optimum conditions such as the ratio of the reactants, pH, and the temperature of the preparation. Furthermore, this study demonstrated the promising results of the applications of these nanoparticles in detecting mercury ions to mitigate their harmful effects on the environment, in addition to their ability to act as an alternative to antimicrobial and anticancer drugs.

The *Haloxylon ammodendron* plant is a big sandy shrub in the Qassim desert in Saudi Arabia, and its scientific name is known as a C4 plant because it uses the C4 carbonization pathway to enhance photosynthetic rate by limiting or decreasing photorespiration [34,35]. The *Haloxylon ammodendron* (HM) shrub has a variety of uses, such as halting sand creep and lowering carbon emissions, pollution, soil contamination, etc. [36,37].

The common chemical ingredients in *Haloxylon ammodendron* shrub were discovered and reported [38–40]. Scheme 1 illustrates the main chemical compounds of HM shrub.

| β-Sitosterol | Stigmasta-5,22-dien-3-ol | Spinasterol | Quercetin |

| Daucosterol | Chikusetsusaponin IVa | Quercetin-3- O - β -D-galactoside | Quercetin-3-O-β-D-glucoside |

**Scheme 1.** The main chemical compounds of the HM shrub.

## 2. Results and Discussion

### 2.1. Mechanism of Formation of Silver Nanoparticles

When talking about the mechanism of the formation of silver nanoparticles, the two correlated and important processes here are the oxidation and reduction between the silver nitrate chemical and the biological solution (HME) [41]. The proper conditions for the reaction, such as the pH, the volume ratio of the reactants, and the reaction temperature, must be provided for the synthesis of silver nanoparticles [42]. These factors affect the size and shape of the nanoparticles formed [43]. At the beginning of the reaction, the ions of the chemical silver nitrate dissociate into the positive silver ion and the negative nitrate ion as a first step [44]. Then, the process of reducing $Ag^+$ to a neutral silver atom, $Ag^0$, through the use of an ethanolic extract from the leaves [45].

In this step, the biologically active compounds (such as quercetin, which contains hydroxyl groups (-OH)) bond to the silver atom and form the nuclei [46]. Then, the particles accumulate on the nuclei due to the electrostatic attraction between them [47]. Temperature increases the reaction kinetics of the ions in the reaction medium and greatly accelerates the reduction process, resulting in the formation of $Ag_2O$ as a reaction product [48]. When the $Ag^+$ group reacts with the negative ions OH, Ag-OH is formed, which is extremely unstable and easily oxidizes to $Ag_2O$ when dried at 50 °C.

At the end of the reaction, Ag-NPs and $Ag_2O$-NPs may be formed in varying proportions, depending on the factors and conditions accompanying the reaction [49,50].

### 2.2. UV-Spectroscopy Analysis

In order to control the formation and stability of silver nanoparticles using the HME, the UV-Vis absorption spectra of the biosynthesized silver nanoparticles were recorded and analyzed. The UV-Vis spectra state the formation of silver nanoparticles as well as the effects of various factors (volume ratio, pH, and temperature) on this formation [51]. The formation of silver nanoparticles was observed by changing the solution color from light yellow to yellowish-brown to deep brown as the factors changed, which was caused by the vibrational excitation of surface plasmon in the silver nanoparticles [52]. A UV-Vis spectrometer detected the surface plasmon resonance (SPR) of Ag-NPs at 413–418 nm. For the biosynthesis of silver nanoparticles, this wavelength value of maximum absorption is consistent with the values measured in previous studies [53]. It depends on the concentrations of silver salt and the functional groups in the plant used in the reaction [54–56].

#### 2.2.1. The Reactant Volume Ratio

When compared to the pure extract, Figure 1 shows the confirmation of the formation of silver nanoparticles. The formation of Ag-NPs was monitored in UV-Vis absorption spectra after different volumes of $AgNO_3$, ranging from 20 mL to 80 mL, were added to a 20 mL HME. The absorbance peaks were shifted to a red wavelength and ranged from 414 nm to 418 nm. This indicates that the mean diameter of Ag-NPs has increased [57–59]. The sample V3, which had a volume ratio of 20 mL to 40 mL of reactants, had the highest absorption. This was the optimal volume ratio for the biosynthesis of Ag-NPs that was determined.

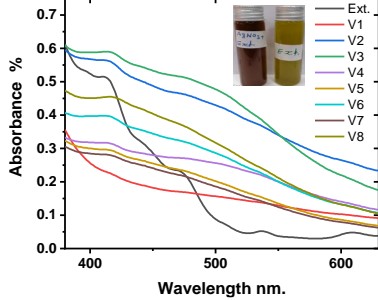

**Figure 1.** The UV-Vis absorption spectra of Ag-NPs at different volume ratios.

### 2.2.2. The pH Factor

Figure 2 demonstrates the effect of pH on the formation of Ag-NPs. It is clear that when changing pH from 3 to 10, the UV-Vis absorbance peak increases and then decreases. The best absorption is for sample p3, pH = 7.5, and this means the probability of silver nanoparticle formation increases in the neutral reaction medium. As well as the absorption peak, which is related to the size and shape of the formed Ag-NPs, the role of the pH factor is to increase and alter the electrical charges of biomolecules, which in turn affect the capping, stability, and growth of Ag-NPs [56,60]. This study demonstrated that the prediction of Ag-NPs formation in the basic bio-reaction is greater than that in the acidic bio-reaction.

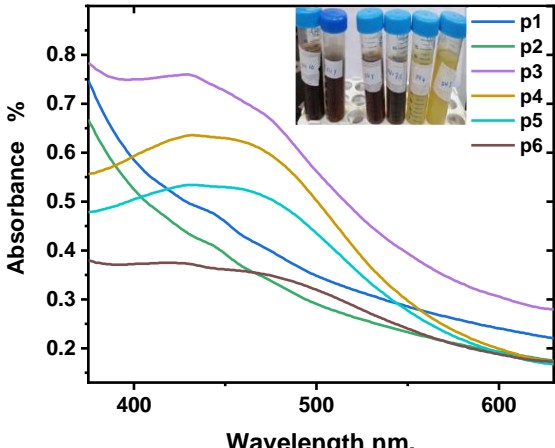

**Figure 2.** UV-Vis absorption spectra of Ag-NPs at various pH levels.

### 2.2.3. The Reaction Temperature

From Figure 3, it is shown that the highest temperature has the highest absorbance. This set of experiments proposes that the high reaction temperature of Ag-NPs biosynthesis will accelerate the formation and growth of Ag-NPs more than at room temperature. This can be explained by the acceleration of the $Ag^+$ reduction rate and then the regular nucleation of silver seed [32,61,62]. The maximum absorbance at high temperatures in sample T3 indicates that the Ag-NPs are smaller due to their formation at shorter wavelengths than at low temperatures.

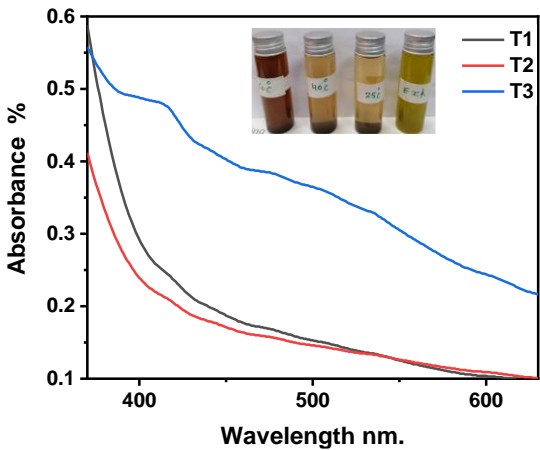

**Figure 3.** The UV-Vis absorption spectra of Ag-NPs at different temperatures.

### 2.3. X-ray Diffraction Patters Analysis

X-ray diffraction is a good and effective technique for verifying the identity of the expected elements present in the prepared nanoparticle samples [63,64]. Figure 4 shows the

XRD pattern of the prepared sample. The positions of the peaks indicate the arrangement of the elements' atoms inside the samples [65]. By comparing these positions of peaks with standard XRD cards, it is shown that there are two phases of crystallinity, Ag and $Ag_2O$, respectively. The peaks at 28.17°, 32.62°, 46.57°, 55.25°, and 57.76° correspond to the planes (100), (110), (200), (220), and (211) as standard data for $Ag_2O$ (JCPDS No. 76-1393). The other peaks at 38.53° and 77.27° correspond to the planes (111) and (311) as standard data for Ag (JCPDS No. 04-0783) [66–68]. The unit cell of the $Ag_2O$ nanoparticles, which are the dominant particles, is cubic (a = 0.476 nm).

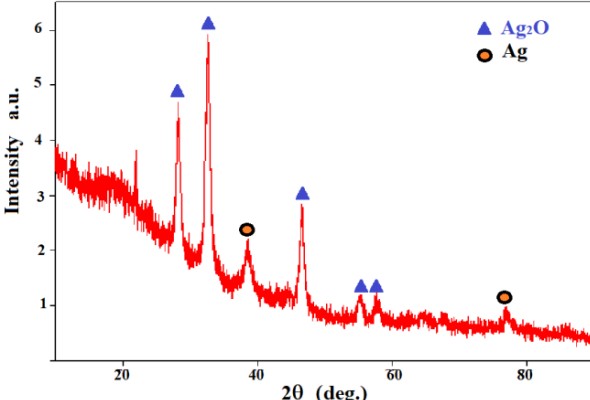

**Figure 4.** XRD pattern of silver nanoparticles.

The crystal size of $Ag_2O$ nanoparticles was computed using the Scherrer formula [69]:

$$D = k\lambda / \beta \cos\theta \tag{1}$$

where k is a constant (~0.94), $\lambda$ is the wavelength of the diffracted X-ray, $\beta$ is the width of half of the maximum diffracted X-ray intensity, and $\theta$ is the angle of the diffracted X-ray. The crystal size of $Ag_2O$ nanoparticles at (110) was 10.64 nm.

### 2.4. SEM Analysis

In Figure 5, the SEM micrograph illustrates the silver nanoparticles with their white color and small spherical particles. It was also clear from the SEM image that the capping agent provided stabilization for silver nanoparticles, caused them to spread out within the aggregates, and prevented them from gathering [70]. The size of these particles was calculated using ImageJ software, and their size distribution is shown in Figure 5b, with an average particle size of 10.97 nm. This value of average particle size is approximately similar to the value of crystal size of these silver nanoparticles from XRD analysis.

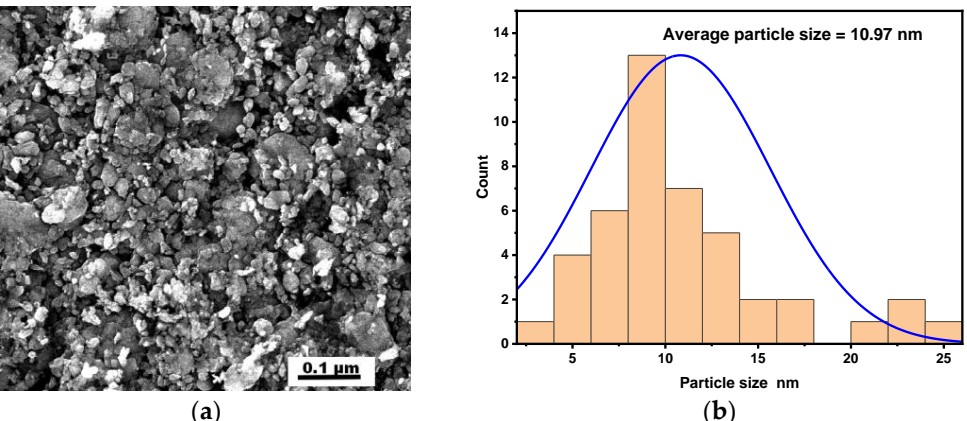

(**a**)  (**b**)

**Figure 5.** (**a**,**b**) SEM image and size distribution of the silver nanoparticles.

### 2.5. FTIR Analysis

Infrared is a useful method for detecting the structural features of unknown substances by identifying the functional groups that absorb the IR energy. In Figure 6, the FT-IR spectra of the pure HME extract and the resultant silver nanoparticle solution are shown. It appears several bands, which represent various functional groups, exist in the chemical compounds available in the HME extract and the solution of silver nanoparticles. The main observation in the functional part of IR is that the broadband absorption of O-H at 3225 cm$^{-1}$ in the IR spectra of HME extract became less peaking and broad and was shifted to 3280 cm$^{-1}$ for the silver nanoparticles' solution. This indicates a reduction in the intense intermolecular hydrogen bonds used in the reduction process to form silver nanoparticles. Furthermore, the difference in mass between the biological extract (HME) molecules and the resultant silver nanoparticles in the O-H absorption bands caused that shift in the wavenumber between them [71,72].

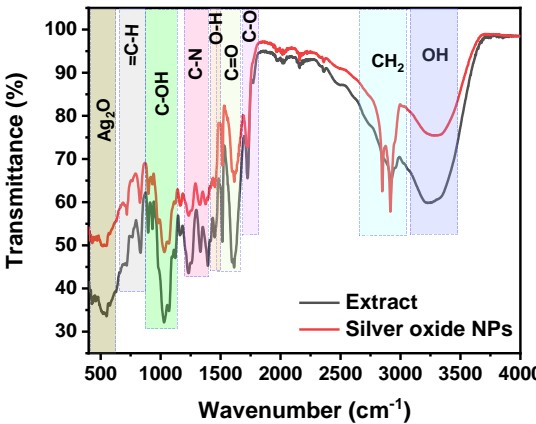

**Figure 6.** FTIR measurements of HME and silver oxide nanoparticles.

On the other hand, the increased peak intensity in the IR spectrum of the resultant silver nanoparticles' solution in the C-H absorption band is greater than that in the HME solution, which asserts that these functional groups also contributed to the reduction process [73].

In the fingerprint part of the IR spectrum, the peak assignment at 520 cm$^{-1}$ is attributed to Ag-O stretch vibration groups, and this value is in good agreement with reported research [71,74]. The other absorption peaks that correspond to the functional groups that are available in the HME solution and contributed to the capping of the silver and silver oxide nanoparticles are shown in Table 1.

**Table 1.** The functional groups' absorption band identification for the HME extract and the silver/silver oxide nanoparticles' solution.

| Functional Group | HME | Silver/Silver Oxide Nanoparticles |
|---|---|---|
| | Peak Assignment (cm$^{-1}$) | |
| O-H stretching vibration [50] | 3222 | 3293 |
| C-H stretching vibration [75,76] | 2847 and 2918 | 2848 and 2916 |
| C-O stretching vibration [77] | 1725 | 1727 |
| C=O stretching vibration [78,79] | 1514 and 1613 | 1515 and 1615 |
| O-H curvature vibration [80] | 1448 | 1454 |
| C-N stretching vibration [81,82] | 1393 and 1232 | 1375 and 1233 |
| C-OH stretching vibration [83,84] | 1029, 898, and 932 | 1033, 898, and 930 |
| =C-H bending vibration [85] | 716, 830 | 718, 828 |

### 2.6. Mercury Ions (Hg$^{2+}$) Sensing Analysis

By looking at all the chemical compounds in the HME, it is clear that all the compounds are polyunsaturated rings with a few unsaturated bonds. These compounds do not contain fluorescent components, with the exception of quercetin compounds.

From Figure 7, the HME solution showed an emission spectrum with a peak at 493 nm and an absorption spectrum with peaks at 243 and 277 nm. These peaks are due to spin-allowed S0 → S2 electronic transitions. Two additional but lower intensity peaks are also observed at 329 nm and 416 nm for the absorption spectra, which indicate an S0 → S1 type electron transition that is spin-allowed [86]. A high Stokes shift between the absorption and emission spectra of about 216 nm is observed for the product solution (HME and Ag nanoparticles), which indicates a large increase in the dipole moment and an energy transfer through non-radiative processes, as shown in Figure 7. The spectra show that there are a variety of functional groups that can bind to metallic elements to create nanoparticles, as undertaken in this study [87–90].

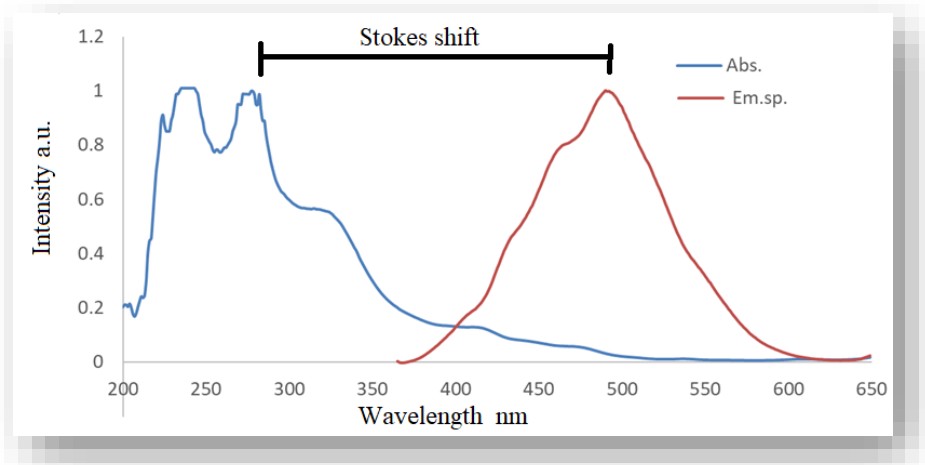

**Figure 7.** The absorption and emission spectra of the HME.

It was found to be a 100% increase in the fluorescence intensity in the product solution (HME and Ag nanoparticles) compared to the pure HME solution, as in Figure 8a. Additionally, a hypochromic shift in the emission maximum spectra of the HME solution of about 20 nm is observed as a result of the presence of silver nanoparticles compared with the pure HgME solution, as in Figure 8b.

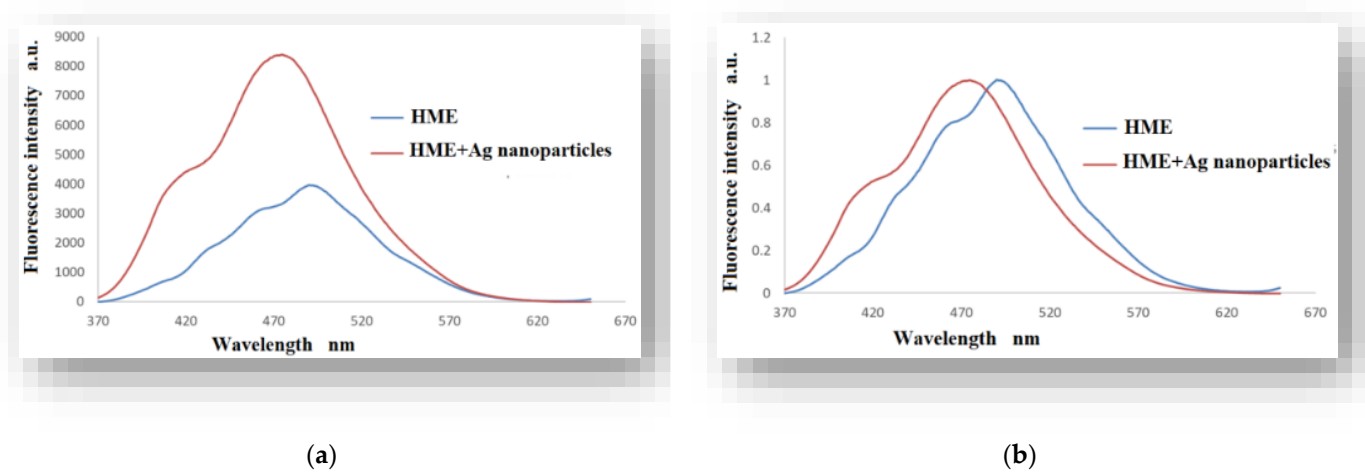

(**a**)          (**b**)

**Figure 8.** (**a**) Fluorescence spectra of the HME and the product solution. (**b**) Normalized fluorescence spectra of the HME and the product solution, $\lambda_{ex}$ = 360 nm.

Figure 9 shows a major quenching in the fluorescence intensity in the presence of $Hg^{2+}$ ions. By measuring the fluorescence intensity when mixing a fixed concentration of the nanoparticle solution (HME and $Ag/Ag_2O$ nanoparticles) with the $Hg^{2+}$ solution, which is at various concentrations. The $Hg^{2+}$ detection capability of the nanoparticle solution was assessed. Fluorescence intensity at 479 nm decreased consistently with increasing $Hg^{2+}$ concentrations when compared to the fresh sample, which was free of $Hg^{2+}$, indicating a clear inverse relationship between fluorescence intensity and $Hg^{2+}$ quantities.

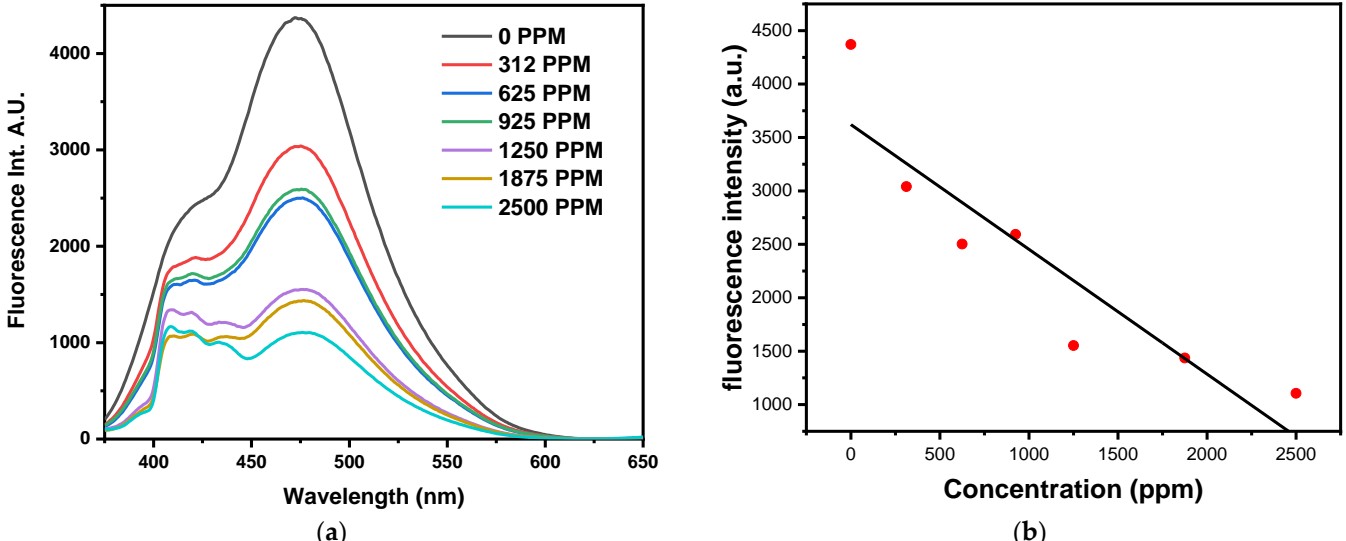

(a)　　　　　　　　　　　　　　　　　　　(b)

**Figure 9.** (**a**,**b**): Fluorescence intensity of the product solution (HME and silver. nanoparticles) with different concentrations of $Hg^{2+}$.

From Figure 9, it is clear that the intensity of fluorescence decreases with the increase in the concentration of mercury ions, and this indicates the existence of an electrostatic interaction between mercury ions and silver nanoparticles, as mentioned in previous research. The potential reaction mechanism depends on the oxidation of mercury ions to the capping (hydroxyl) of the silver nanoparticles. This leads to the transformation of mercury ions into mercury particles (surrounded by hydroxyl). Next, a part of the mercury ions interacts with silver ions and turns into a silver–mercury molecule, and then gradually, the silver nanoparticles turn into normal silver particles, and the concentration of the fluorescence intensity of the product decreases [23,49,91].

Figure 10 shows the Stern–Volmer plot derived from the following equation of silver nanoparticle fluorescence quenching using $Hg^{2+}$ as a quencher [92]:

$$I_0 / I = 1 + K_{sv} \, [Q] \tag{2}$$

where [Q] is the quencher concentration $[Hg^{2+}]$, I and $I_0$ are the fluorescence intensities of studied quercetin Ag nanoparticles in the presence and absence of $Hg^{2+}$, respectively, and KSV is the Stern–Volmer quenching constant.

In Figure 10, the system is said to follow the Stern–Volmer relationship. The value of the Stern–Volmer quenching constant was calculated from the slope of the linear fitting of plotting $I_0 / I$ vs. $[Hg^{2+}]$ concentration and is equal to $2.3 \times 10^{11} \ M^{-1}$, which is much higher than the diffusion control of HME, indicating that the mechanism of fluorescence quenching can be called a dynamic diffusion process [93].

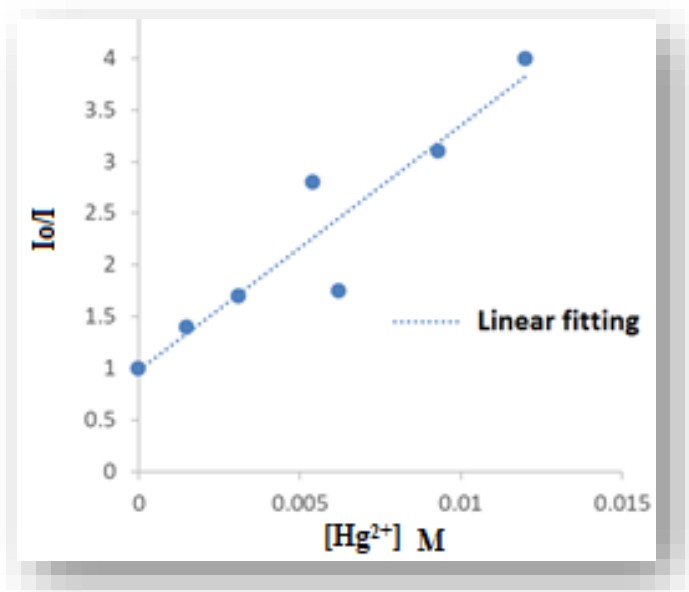

**Figure 10.** The Stern–Volmer plot for fluorescence intensity vs. $Hg^{2+}$ concentration.

*2.7. Antibacterial and Antifungal Activities*

The antibacterial and antifungal activities of the HME extract, AgNPs, ampicillin, and colitrimazole against Gram-positive bacteria, Gram-negative bacteria, and two common strains of fungi, respectively, were tested, and the results are listed in Table 2.

**Table 2.** The antibacterial and antifungal activities of the HME extract, AgNPs, ampicillin, and colitrimazole.

| Microbes Name | Zone of Inhibition in mm | | | |
| --- | --- | --- | --- | --- |
| | **HME Extract** | **AgNps** | **Ampicillin** | **Colitrimazole** |
| *E. coli* | 13.21 | 19.03 | 23.55 | ---- |
| *Pseudomonas aeuroginosa* | 10.50 | 17.02 | 34.46 | ---- |
| *S. aureus* | 11.14 | 18.20 | 25.47 | --- |
| *Bacillus subtilis* | 12.56 | 18.24 | 32.81 | --- |
| *C. Albicans* | 14.16 | 17.34 | --- | 23.75 |
| *A. flavus* | 12.27 | 16.14 | ---- | 25.38 |

The HME extract and AgNPs show a medium effect in inhibiting all kinds of bacteria and fungi, while the standard antibiotics show maximum inhibition for some kinds of microbes and noninhibition for other kinds (fungi). The second finding is that HME extract [94] enhances the antimicrobial activity of AgNPs [95], which explains why AgNPs' antibacterial and antifungal activities are more effective than HME extract. Additionally, the outcomes demonstrated that silver nanoparticles are the most powerful inhibitors of all microbial life. This antibacterial activity of silver nanoparticles shows their potent capacity to halt the spread of numerous types of microbes and eradicate them. This antimicrobial activity was explained in numerous studies through different hypotheses, which have not been proven until now [29].

*2.8. Anticancer Activity*

Assays for cell proliferation are used to measure the number of cells over time, metabolic activity, and the frequency of cell division. Cytotoxicity, cell viability, or prolifera-

tion can all be assessed with the MTT (3-(4,5-dimethylthiazol-2-yl)-2,5-diphenyltetrazolium bromide) assay.

Generally, this MTT assay is an accepted paradigm for determining the complexes' antitumor efficacy and also enables the assessment of metabolic cell activity [96]. The cytotoxicity of the HME extract and AgNPs against a human lung fibroblast line (WI38), hepatocellular carcinoma (HEPG-2), and breast cancer cell line of the mammary gland (MCF-7) was evaluated using different concentrations of compounds. The AgNPs exhibit potent anticancer properties and possess severe toxicity towards the tested cancer cell type. Table 3 provides the IC50 values for the HME extract and AgNPs.

**Table 3.** The anticancer activity of the HME extract, AgNPs, and doxorubicin against WI38, HEPG-2, and MCF-7 cells.

| Comp. Name | * (IC$_{50}$ (µM)) | | |
|:---:|:---:|:---:|:---:|
| | **WI38** | **HEPG-2** | **MCF-7** |
| HME | $33.02 \pm 3.9$ | $27.15 \pm 2.1$ | $37.04 \pm 2.2$ |
| AgNPs | $14.07 \pm 2.5$ | $15.58 \pm 3.6$ | $13.39 \pm 3.4$ |
| DOX | $6.72 \pm 0.5$ | $4.50 \pm 0.2$ | $4.17 \pm 0.2$ |

* IC50 (µM): from 1 to 10 is very strong, from 11 to 20 is strong, from 21 to 50 is moderate, and from 51 to 100 is weak. Above 100 (non-cytotoxic), and DOX means Doxorubicin.

The results revealed that the potency anticancer activity of HME extract was greater than that of silver nanoparticles or doxorubicin. In Table 3, the silver nanoparticles have more inhibitory activity in HEPG-2 cells.

The effect of reactive oxygen species (ROS) can be used to interpret the reaction mechanism of silver nanoparticles with cancer cells. The production of ROS is due to the biosynthesis of AgNPs using plants rich in OH groups. These silver nanoparticles react with the cell wall to form peroxide ($H_2O_2$) and change the function of the mitochondria by increasing the adenosine triphosphate (ATP) level, uncoupling the respiration process, ultimately causing cellular death [97–99].

## 3. Materials and Instruments

### 3.1. Materials

The *Haloxylon ammodendron* plant was collected from the Qassim desert in Saudi Arabia. Silver nitrate, nitric acid, ammonia, absolute ethanol, and mercury chloride were supplied by Sigma-Aldrich, deionized water from Qassim laboratory.

### 3.2. The Haloxylon ammodendron Ethanolic Extract Preparation

The *Haloxylon ammodendron* leaves were collected from the Qassim area, Saudi Arabia, during the springtime of 2023.

The *Haloxylon ammodendron* (HME) was washed several times with distilled water and once with deionized water. After drying out in the shade in the laboratory for five days, the *Haloxylon ammodendron* plant was ground into a fine powder. By soaking 100 g of plant powder in 300 mL of 100% ethanol for two days at room temperature using a magnetic stirrer, samples of the *Haloxylon ammodendron* plant were extracted. After that, the mixture was filtered. The ethanol extract of the *Haloxylon ammodendron* (HME) solution was cooled and stored in sterile vials until use.

### 3.3. The Biosynthesis of Ag Nanoparticles

In the preparation of the empirical tests, 0.001 mM of Ag(NO$_3$) and HME were both used to achieve the best preparation of Ag nanoparticles, which was the result of several attempts. The effects of reactant volume ratio, pH, and reaction temperature on the biosynthesis of Ag nanoparticles were studied as follows:

### 3.3.1. The Reactant Volume Ratio

In this part, many experiments were conducted to determine the best volumetric ratio of the reactants by mixing a fixed amount of HME (20 mL) with different volumes of silver nitrate (20, 30, 40, 50, 60, 70, 80, and 90 mL). The samples were labeled with V1, V2, V3, V4, V5, V6, V7, and V8, respectively. The mixing process was carried out at room temperature with continuous stirring for 3 h. During each experiment, the mixture color changes from greenish-yellow to deep brown, and a precipitate forms, which indicates the formation of silver nanoparticles. This precipitate of silver nanoparticles was extracted using a centrifuge and then dried at 50 °C in an open oven for 8 h. Finally, after drying, the samples were prepared for X-ray diffraction and UV-vis absorption spectroscopy measurements.

### 3.3.2. The pH

In order to study the effect of pH on the biosynthesis of silver nanoparticles and determine the best reaction environment for the formation of these particles, several experiments were conducted. In these experiments, the volume ratio of the reactants (silver nitrate and HME) was fixed based on the best result of the experiments in the previous part. The pH of the mixture was adjusted by using drops of both nitric acid solution and ammonia solution. Samples were obtained at different pH values (3, 4, 7.5, 8, 9, and 10) while keeping the same previous preparation conditions, such as the temperature and speed of stirring the mixture and molar ratio. The samples were named p1, p2, p3, p4, p5, and p6, respectively. The same previous steps were followed in extracting the silver nanoparticle precipitate, and the same measurements were made.

### 3.3.3. The Reaction Temperature

To find out the effect of reaction temperature on the biosynthesis of silver nanoparticles and to obtain the suitable and optimal reaction temperature for the formation of these particles, experiments were conducted. In these experiments, the volume ratio of the reactants and pH were fixed based on the best results from the earlier experiments in the previous parts. The mixture was heated at different temperatures (25, 40, and 60 °C), and the resulting samples were labeled T1, T2, and T3, respectively. The formed silver nanoparticle precipitate was extracted using the same previous steps, and then XRD and UV-Vis absorption spectra were measured.

### 3.4. Characterization Instrumentation and Techniques

The pH values in all experiments were measured using a JENWAY 3510 pH meter from the UK.

An Agilent spectrometer (Cary 600 FTIR, USA) was used to analyze the Fourier-transform infrared (FTIR) spectra in the solid state. It was conducted in the 4000–400 $cm^{-1}$ wavenumber range.

Using scanning electron microscopy-SEM (JOEL Jsm-5500), the size and shape of biosynthesized silver nanoparticles were examined. For the size investigation of diverse samples, a 30 kV accelerating voltage and various magnification techniques were used.

A Shimadzu spectrophotometer (UV-1650PC, Japan) with a 1 cm quartz cell and a wavelength range of 250–650 nm was applied to study the UV–Vis spectra of the HME and the AgNPs.

For fluorescence measurements, seven samples of $Hg^{2+}$ ions were prepared at different concentrations (0, 312, 625, 925, 1250, 18750, and 2500 ppm) using DI water and $HgCl_2$. A required amount of silver nanoparticles' solution, which was biosynthesized using *Haloxylon ammodendron*, was added to the $Hg^{2+}$ samples and monitored by the Jasco fluorescence spectrometer FP-8200.

A Rigaku XRD diffractometer (Ultima IV, USA) with Cu-K ($\lambda$ = 0.15418 nm) as anode material was employed to study the nanostructure of the Ag nanoparticles. The XRD system was operated at 30 mA and 40 kV.

### 3.5. In Vitro Antimicrobial Studies

The biological activity of the extraction and its AgNps was investigated in vitro as an antimicrobial against some different types of bacteria and fungi. It was tested on four pathogenic bacterial strains (*Bacillus subtilis*, *Staphylococcus aureus*, *Pseudomonas aeruginosa*, and *Escherichia coli*) and two types of fungi (Candida albicans and Aspergillus flavus) using the agar-well diffusion method [31]. In this method, the microbial cells (bacteria or fungi) were inoculated on Petri dishes, and the test samples (HM extraction and AgNps) were implanted inside the holes, which were made in the middle of these microbial cell dishes. Afterward, these dishes were incubated for different hours depending on the microbe's type (18–24 h at 37 °C), and thereafter, the diffusion of microbes was determined by measuring the zone of inhibition. The presence of antimicrobial activity is indicated by the absence of microbe growth near the test samples. The antimicrobial activity test process was duplicated three times using the same concentration. The antibiotics ampicillin and clotrimazole were used as standard antibiotics against bacteria and fungi, respectively.

### 3.6. Anticancer Activity

The viability assay was used to evaluate cytotoxicity, and the cancer cells (hepatocellular carcinoma (HEPG-2), human lung fibroblast (WI38), and mammary gland breast cancer (MCF-7) were used to study the anticancer activity of the samples. As well, doxorubicin was used as a standard anticancer drug for comparison [96].

## 4. Conclusions

This work demonstrated the viability of synthesizing silver nanoparticles from the Saudi *Haloxylon ammodendron* shrub and identified the ideal parameters for their manufacture (the reactant volume ratios, pH, and temperature). The preparation conditions helped form two types of silver nanoparticles due to the presence of oxygen gas during drying. These two types of nanoparticles are silver oxide nanoparticles and silver nanoparticles. Through X-ray diffraction analysis, it turned out that the formation percentage of silver oxide nanoparticles is greater than that of silver nanoparticles. The results of infrared and ultraviolet spectroscopy of the materials resulting from the interaction of silver nitrate salt with the HME extract confirmed the formation of silver nanoparticles (whether silver oxide or silver) due to the presence of oxidizing and active substances in the HME extract. The microscopic images obtained using the scanning electron microscopy technique showed the spherical shape of these particles and that the formed nanoparticles accumulate in large numbers, which explains the broadening of the ultraviolet spectrum curves of these nanoparticles. Experiments with fluorescence quenching, which was used in this research as a technique to get rid of mercury ions in aqueous solutions, showed the high ability of these nanoparticles to remove these heavy ionic pollutants.

The HME extract showed promising results as an antibacterial, antifungal, and anticancer material. The use of HME extract in the biosynthesis of silver nanoparticles enhanced the biological activity of these nanoparticles and made them more effective at inhibiting different microbes and cancer cells.

**Author Contributions:** Conceptualization, A.N.A.-H. and T.M.A.; methodology, A.N.A.-H. and R.A.A.; validation, A.N.A.-H., T.M.A. and R.A.A.; formal analysis, A.N.A.-H. and R.A.A.; investigation, R.A.A. and A.N.A.-H.; resources, A.N.A.-H.; data curation, A.N.A.-H.; writing—original draft preparation, A.N.A.-H. and R.A.A.; writing—review and editing, A.N.A.-H. and T.M.A.; visualization, A.N.A.-H.; supervision, A.N.A.-H.; project administration, A.N.A.-H.; funding acquisition, R.A.A. and A.N.A.-H. All authors have read and agreed to the published version of the manuscript.

**Funding:** This research was funded by "Qassim University", represented by the deanship of scientific research, under the number (COS-2022-1-1-J-24993) during the academic year 1444 A H/2022 A D.

**Acknowledgments:** The authors gratefully acknowledge "Qassim University", represented by the deanship of scientific research, on the financial support for this research under the number (COS-2022-1-1-J-24993) during the academic year 1444 A H/2022 A D.

**Conflicts of Interest:** The authors confirm that they do not have any conflict of interest related to the work in a manuscript.

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
