# Peer review of "The Effect of the Saudi Haloxylon ammodendron Shrub on Silver Nanoparticles: Optimal Biosynthesis, Characterization, Removability of Mercury Ions, Antimicrobial and Anticancer Activities"

_inorganics, doi:10.3390/inorganics11060246_

Round 1

Reviewer 1 Report

The manuscript reports a well-developed study. However the text has problems to be solved. Besides a language problem, many of the sentences in the text are too long and confusing.

The purpose of the work is not focused and well described.

The materials and methods section and the discussion of the results need to be expanded and more detailed

Comments to improve the quality of the manuscript are given directly on the pdf version of the manuscript

Author Response

dear reviewer, thanks a lot for reviewing my manuscript   

Dear Reviewer No. 1:
We welcome your valuable advice and comments that helped us enhance the presentation of our manuscript. Most of your suggestions have been implemented. You can find below our point-by-point reply to your comments, and we think this is sufficient.

Comments and Suggestions for Authors

The manuscript reports a well-developed study. However the text has problems to be solved. Besides a language problem, many of the sentences in the text are too long and confusing.

The purpose of the work is not focused and well described.

Thank you very much for this comment. We have modified the introduction for this purpose and made it clearer and focused.

The materials and methods section and the discussion of the results need to be expanded and more detailed

We welcome this advice, and we have added new data and results as well as more discussion and marked them in red.

Comments to improve the quality of the manuscript are given directly on the pdf version of the manuscript

Thank you again. We have modified and revised the manuscript as per your suggestion in the PDF file.

Reviewer 2 Report

The manuscript synthesized biomass silver nanoparticles using Saudi Haloxylon ammodendron shrubs and investigated their ability to remove mercury ions. It should be emphasized that HME is not a raw material for the preparation of silver, but an auxiliary reduction of silver ions. Major revisions are required before it could be considered for publishing.

1. The diffraction peak data of the standard XRD cards should be supplemented in Fig. 4. The grain size calculated by Scherrer's formula is not accurate, and the specific data and calculation process are not given.

2.SEM lacks scale bars and the images are fuzzy, which makes it difficult to obtain effective material morphological features.

3. The font format in the chart should be uniform. Some of the graphs are so poorly handled that they don't make sense.

4. Improve the details of characterization analysis.

5. The study is too shallow, without in-depth analysis, and it is not clear what scientific issue the authors are trying to illustrate.

6. The writing logic is confused and the experimental scheme design is unreasonable.

7. Abstract and title are misleading to readers and need to be changed.

8. In addition, the diagramming of the manuscript is arbitrary.

9. Waste water treatment is an interesting field. Many good researches have been done. Some typical references are suggested to be cited, e.g. Journal of Bioresources and Bioproducts 2022, 7 (2), 109-115; Journal of Bioresources and Bioproducts 2021, 6 (4), 292-322.

Minor polishing and editing of English language required.

Author Response

dear reviewer, thanks a lot for reviewing my manuscript

Dear Reviewer No. 2:
We appreciate you taking the time to read this manuscript and providing us with the available feedback to make our manuscript better. We've tried to implement the majority of your comments. You will find below our response to your comments, point by point, and we hope it is sufficient.

We have applied most of these points in the manuscript and marked in red.

Comments and Suggestions for Authors

The manuscript synthesized biomass silver nanoparticles using Saudi Haloxylon ammodendron shrubs and investigated their ability to remove mercury ions. It should be emphasized that HME is not a raw material for the preparation of silver, but an auxiliary reduction of silver ions. Major revisions are required before it could be considered for publishing.

We agree with you, and we have changed the title of the manuscript accordingly. 

  1. The diffraction peak data of the standard XRD cards should be supplemented in Fig. 4. The grain size calculated by Scherrer's formula is not accurate, and the specific data and calculation process are not given.

Thank you for this important note, and to make sure about our calculation, we can say that:The diffraction peak data that we obtained from the X-ray diffractometer laboratory were included in this manuscript as a supplementary file. From this file, the full width-half maximum (FWHM) at the scattering angle of 32.624 is 0.773 degrees. Using this value, we have calculated the crystal size.From the supplementary file, you can also find the cards that were used in the XRD analysis, but we have made more surveys to find the most appropriate cards according to the XRD data, which were JCPDS No. 76-1393 and JCPDS No. 04-0783.

2.SEM lacks scale bars and the images are fuzzy, which makes it difficult to obtain effective material morphological features.

We appreciate your comment on this point, and we have retaken a new SEM photo with a scale bar. We have also computed the particle size using ImageJ software and included these additions in the manuscript in red.

  1. The font format in the chart should be uniform. Some of the graphs are so poorly handled that they don't make sense.

We thank you again, and we have changed some of these graphs as well as the font size of this chart to make them uniform.

  1. Improve the details of characterization analysis.

We have added new data and analysis to the manuscript and marked it in red.

  1. The study is too shallow, without in-depth analysis, and it is not clear what scientific issue the authors are trying to illustrate.

Maybe it looked as you described, but we have added new data and results as well as changed the title and the introduction to make the manuscript more clearly oriented and targeted.

  1. The writing logic is confused and the experimental scheme design is unreasonable.

Thank you for your observation, and we have attempted to make some corrections to the experimental section and marked it in red.

  1. Abstract and title are misleading to readers and need to be changed.

Thanks for your honesty, and we have altered both the title and the abstract and marked the changes with red.

  1. In addition, the diagramming of the manuscript is arbitrary.

We do not understand your comment, but we think that we have tried to follow the template of the journal design. However, we are ready for any further modifications.

  1. Waste water treatment is an interesting field. Many good researches have been done. Some typical references are suggested to be cited, e.g. Journal of Bioresources and Bioproducts 2022,7(2), 109-115; Journal of Bioresources and Bioproducts 2021, 6 (4), 292-322.

Thank you for this suggestion, and we have cited these scientific papers in our manuscript.

Comments on the Quality of English Language

Minor polishing and editing of the English language are required.

We have revised the language of the manuscript and made the required changes in red.
